# A comparison of dataset distillation and active learning in text classification

## Abstract

Deep learning has achieved great success over the past few years in different aspects ranging from computer vision to natural language process. However, the huge size of data in deep learning has always been a thorny problem in learning the underlying distribution and tackling various human tasks. To alleviate this problem, knowledge distillation has been proposed to simplify the model, and later dataset distillation as a new method of reducing dataset sizes has been proposed, which aims to synthesize a small number of samples that contain all the information of a very large dataset. Meanwhile, active learning is also an effective method to reduce dataset sizes by only selecting the most significant labeling samples from the original dataset. In this paper, we explore the discrepancies in the principles of dataset distillation and active learning, and evaluate two algorithms on NLP classification dataset: Stanford Sentiment Treebank. The result of the experiment is that the distilled data with the size of 0.1% of the original text data achieves approximately 88% accuracy, while the selected data achieves 52% performance of the original data.

Keywords: knowledge distillation  dataset distillation  active learning  text classification

## 1 Introduction

Deep learning whose neural network has a large number of layers in order to solve complicate human tasks has achieved great success over the past few years in many various applications ranging from computer vision to natural language process. These networks have to be trained on a large number of data so as to learn the underlying distribution of the task. However, the huge computational complexity and the massive storage which deep learning requires are two main setbacks in the study. Therefore, data reduction has become a popular and feasible method to help people to learn the model and solve the problem by a small number of data.

Nowadays, to improve the efficiency of deep neural networks, knowledge distillation (Hinton et al., 2015) has been proposed to compress the complex model into the simpler model and remain the approximately same accuracy as the original model. In knowledge distillation, the teacher model transfers the knowledge to the student model which has the less layers and storage. It asks the student model to achieve two goals: minimizing the loss function between the teacher model's soft prediction (softmax(T=t) i.e. T is the distillation temperature) and student model's soft prediction which means the distilled model need to achieve as much performances as the original model; minimizing the loss function between the student's hard prediction (softmax(T=1)) and the ground truth. It actually plays a role of model compression in deep learning which can reduce over-fitting and is widely used in data mining of infinitely large, unsupervised datasets; less or zero sample learning; migration learning and etc.

Different from knowledge distillation which transfers the complex model to the simpler model, data distillation (Wang et al., 2018) is targeted to encapsulate all the knowledge from a large number of data to a small number of data. It is an alternative formulation to reduce the data sizes by synthesizing a small number of data points containing all the

information of a large dataset that do not need to come from the correct true dataset distribution. Compared with other methods which aim to reduce the size of dataset such as prototype (Garcia et al., 2012) selection in nearest-neighbor classification, dataset distillation aims to generate a small number of datasets from the original data rather than selecting samples from the true distribution. It depends on backpropagated gradient to compress a dataset into a small set of synthetic data samples which is good for a complex neural network to be trained on a small number of data rather than the original massive dataset with the almost similar accuracy.

Sometimes, we can regard deep learning as a situation that students always learn everything in detail from teachers in class while the amount of knowledge is very huge, however active learning is an effective way to learn parts of knowledge actively which is selected deliberately to master the knowledge to the maximum extent. It is a cyclic procedure between a teacher or an oracle (usually a human annotator) and an active learner (Schröder & Niekler, 2020). Compared with passive learning which is only to feed the data to the model, active learning choose which samples to be labeled by a human expert, the so-called human in the loop.

In this paper, we discuss various discrepancies between two methods of reducing dataset sizes and use two algorithms to solve text classification on NLP classification dataset: Stanford Sentiment Treebank (SST) which consists of 21,5154 phrases from movies with fine-grained sentiment labels in the range of 0 to 1. We use the classification accuracy to compare the performances of active learning and dataset distillation on text classification.

## 2   Related Work

### 2.1   Knowledge Distillation

Hinton et al. have proposed knowledge distillation as a method for imbuing smaller, more efficient networks with all the knowledge of their larger counterparts (Hinton et al., 2015). And then knowledge distillation has been used in various aspects. Linfeng Zhang et al. have proposed self distillation, which notably enhances the accuracy of convolutional neural networks through shrinking the size of the network rather than aggrandizing it (Zhang et al., 2019) .To mitigate the massive number of parameters in deep neural networks, Sukmin Yun et al. have proposed a new regularization method that penalizes the predictive distribution between similar samples and distill the predictive distribution between different samples of the same label during training (Yun et al., 2020) . For casual distillation in language models, Zhengxuan Wu et al. have proposed to augment distillation with a third objective that encourages the student to imitate the causal computation process of the teacher through interchange intervention training (Wu et al., 2021) .

### 2.2   Data Distillation

Data distillation has been proposed firstly by Tongzhou Wang as an alternative formulation to synthesize a small number of data from the massive original dataset (Wang et al., 2018) . It has been used in various industries. Tian Dong et al. have explored that dataset distillation is a better solution to replace the traditional data generators for private data generation because the distilled data is a set of synthetic data which will not disclose the original information (Dong et al., 2022). Shengyuan Hu et al. have proposed a new scheme which includes gradient compression via dataset distillation in federated learning for upstream communication where every client learners use a light-weight distilled dataset as the training data instead of transmitting the model update (Hu et al., 2022) . To reduce catastrophic forgetting in deep neural networks, Wojciech Masarczyk et al. have investigated the use of synthetic data to generate two-step optimisation process via meta-gradients (Masarczyk & Tautkute, 2020). To alleviate the storage and time for training neural models on large-scale graphs in real world applications, Wei Jin et al. have proposed the condensation problem by imitating the GNN training trajectory on the original graph through the optimization of a gradient matching loss and design a strategy to condense node futures and structural information simultaneously (Jin et al., 2021). Yongqi Li et al. have developed a novel data

distillation method for text classification and evaluate their method on eight benchmark datasets (Li & Li, 2021).

## 2.3 Active Learning

In the era of the deep learning explosion, various tasks and applications are considered to be solved using a data-driven learning approach, which places even higher demands on data. In practice, it is not possible to abandon either labelling or unlabeled data altogether, and active learning can provide a more reasonable expedient to label valuable data without labeling it all, selectively. For low-data regimes and on self-trained vision transformers, Amin Parvaneh et al. have proposed a novel method for batch active learning whose performances are especially significant (Parvaneh et al., 2022). To remove label noise by fully re-annotating large datasets is infeasible in resource-constrained settings, Melanie Bernhardt et al. have proposed to use active label cleaning to rank instances and introduce a simulation framework to evaluate relabeling efficacy (Bernhardt et al., 2022). Yuhang Zhang et al. have presented a new active learning framework, which for the first time incorporates contrastive learning into recently proposed one-bit supervision, which is more efficient than previous active learning methods requiring assigning accurate labels to the queried samples (Zhang et al., 2022).

## 3 Methodology

### 3.1 Data Distillation

Data distillation pay more attention on transferring knowledge and information from the original dataset to the distilled data, rather than reconstructing samples from the true distribution. Therefore, we have to consider the connection between the distilled data and the original data, which can be linked with the gradient (Wang et al., 2018). When we use the distilled data to train the model, the weight of the network is from various functions from synthetic data, and backpropagated gradient can be used to track the original data to close the gap between the distilled data and the original data. For text dataset, the original data and the distilled data can be settled, then the distilled data can be seperated into several batches so as to train the model by applying the gradient descent to these batches. While we find the appropriate function which input the distilled data and output the well-trained model, which means that when we divide the dataset into batcher in a proper way then we can use the distilled data to train the neural network as the normla text data.

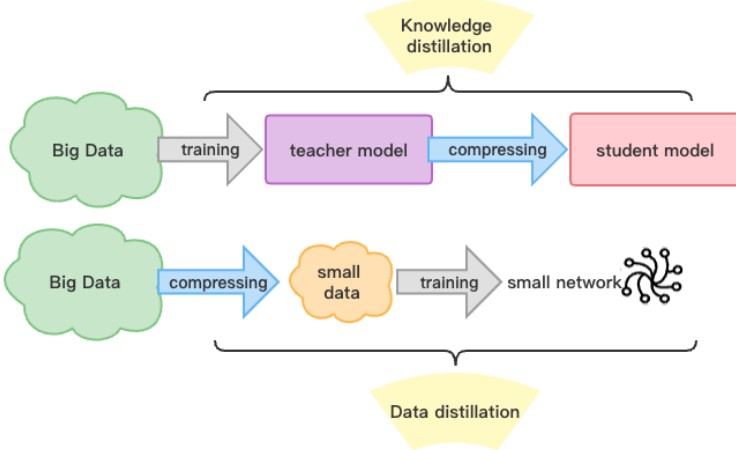

Figure 1: Illustration of knowledge distillation and data distillation

As for data distillation, considering a training dataset $x = \{x_i\}_{i=1}^N$, we parameterize our neural network as $\theta$ and denote $l(x_i, \theta)$ as the loss function that represents the loss of this network on a data point $x_i$. The target of data distillation is to find the minimizer of the empirical error over entire training data:

$$\theta^* = \arg\min_\theta \frac{1}{N} \sum_{i=1}^N l(x_i, \theta) \triangleq \arg\min_\theta l(x, \theta)$$

Applying the data distillation in text classification, considering a text training dataset $T = \{p_i, q_i\}_{i=1}^M$, where $p_i$ is a piece of text and $q_i$ is the corresponding label of the text, and M is the number of samples in original dataset. It is noticeable that the text data is discrete, we may use mathematical matrixes rather than these discrete words to form the distilled data.

The current parameters can be updated with the learning rate $\eta$ and the loss function as follows:

$$\theta_{s+1} = \theta_s - \eta \nabla_{\theta_s} l(\tilde{p}_s, \theta_s)$$

However, it takes a number of update steps in the training process, so we use the data distillation. It means that we aim to learn a small number of synthetic distilled training data then we can apply random gradient descent on every minibatch to find the optimal.

The distilled dataset $\tilde{T} = \{\tilde{p}_i, \tilde{q}_i\}_{i=1}^N$, where $N << M$. Due to the standard training usually applies minibatch stochastic gradient descent or its variants, for text classification, we randomly initialize the distilled data as $\tilde{T}_0$ and divide them into various minibatches $\{\tilde{p}_s\}_{s=1}^{S_1}$, where $S_1$ is the number of batches. Then, when inputing a minibatch of data, the parameter is renewed by the learning rate $\tilde{\eta}$ and the loss function:

$$\theta_1 = \theta_0 - \tilde{\eta} \nabla_{\theta_0} l(\tilde{p}, \theta_0)$$

Therefore, on the basis of the parameters and the initial $\theta_0$, we obtain synthetic data $\tilde{D}_*$ which minimizes the loss function:

$$\tilde{D}_* = \arg\min_{\tilde{D}} l(p, \theta_1) = \arg\min_{\tilde{D}} (p, \theta_0 - \tilde{\eta} \nabla_{\theta_0} l(\tilde{p}, \theta_0))$$

where we derive the new weights 1 as a function of distilled data $\tilde{D}$ and learning rate $\tilde{\eta}$ using Equation 2 and then evaluate the new weights over all the training data $D$. The loss function $l(\tilde{p}, \theta_0)$ is differentiable, and can thus be optimized using standard gradient-based methods.

Afterwards, we can use $\{\tilde{p}_s\}_{s=1}^{S_1}$ to make $\tilde{T}_0$ more close to $\tilde{T}_S$, then $\tilde{T}_S$ can be used as the original data to train the neural network.

### 3.2 Active Learning

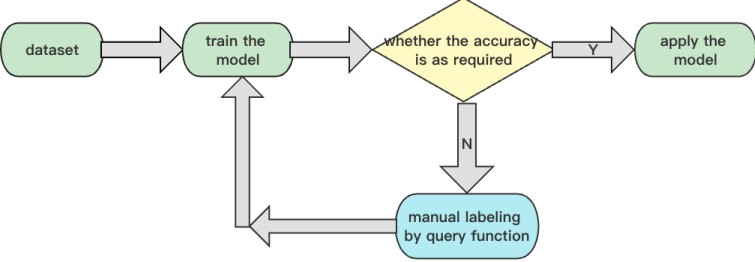

Figure 2: process of active learning

As an effective method for reducing sizes of dataset, active learning is targeted to label the most important data from the overall dataset, which means that it is kind of like choosing the sample from true distribution and reconstructing a life-like sample rather than using the part of original dataset.

As the figure2 illsutrates that the first step of active learning is gathering the data. Afterwards, some data are selected to train the model while at this time, the training model's accuracy is not good enough. Therefore, query functions determine data selection choices of the model in the AL loop. They are used in manual labeling which means that human experts apply the query functions to choose the corresponding label and choose more labeled data to train the model. When the accuracy is as required, the model can be utilized in real human tasks.

We considered multiple methods of active learning in text classification as follows (Jacobs et al., 2021).

(1) Variation Ratio:

The variation ratio measures dispersion around the class. The below equation shows how the variation ratio iscomputed, where $f_x$ denotes the mode count and $F$ the number of stochastic forward passes.

$$v[x] = 1 - \frac{f_x}{F}$$

Actually, variation ratio is a form of predictive uncertainty rather than capturing the uncertainty in the model.

(2) Predictive Entropy:

In mathematics, entropy can be used to measure the uncertainty of a system, with higher entropy indicating greater uncertainty and lower entropy indicating less uncertainty. Therefore, in a dichotomous or multiclassification scenario, those sample data with higher entropy can be selected as the pending annotation data. Entropy $H(x)$ has been defined as:

$$H(x) = -\sum_{i=1}^{n} p(x_i)log_2 p(x_i)$$

where $p(x_i)$ gives the probability of the $i-th$ possible value for the symbol. Entropy is used to quantify the information of data. We can compute the chance of the model classifying a data point by averaging over the softmax probability distributions across the $T$ stochastic forward passes.

$$H[y|x, D_{train}] = -\sum_{c}(\frac{1}{F}\sum_{f} p(y = c|x, \hat{\omega}_f))log(\frac{1}{F}\sum_{f} p(y = c|x, \hat{\omega}_f))$$

where $\hat{\omega}_f$ denotes the stochastic forward pass $f$, and $c$ the number associated to the class-label.

(3)Bayesian Active Learning by Disagreement:

This is a form of conditional mutual information, the condition or the third variable being the training data $D_{train}$. Houlsby et al. used this form of mutual information in an active learning setting and dubbed it Bayesian active learning by disagreement (BALD) (Houlsby et al., 2011).

$$I[y, \omega|x, D_{train}] = -\sum_{c}(\frac{1}{F}\sum_{f} p(y = c|x, \hat{\omega}_f))log(\frac{1}{F}\sum_{f} p(y = c|x, \hat{\omega}_f))$$
$$-\frac{1}{F}\sum_{c,f} p(y = c|x, \hat{\omega}_f)logp(y = c|x, \hat{\omega}_f)$$

## 4 Experiment

### 4.1 Data

SST consists 11,850 sentences and 5 classes of data in the following way:

Table 1: An overview of SST in the experiment

| Datset | Classes | Train Samples | Test Samples |
|---|---|---|---|
| SST | 5 | 10534 | 1316 |

The reason why we choose SST as the experiment dataset is that its size is massive and it can be a benchmark for language models. It allows for the evaluation of AL and data distillation for a larger dataset and for comparison with results found in related work.

To show the efficiency of the distilled data in text classification, we compare the accuracy between the original data and the distilled data on the same test set. Here, we choose 0.1% of the dataset size applying in the distilled data and random samples.

As is known to us all, the most important factor in active learning is to choose an appropriate query function to solve the task in a required accuracy. In this paper, we use various query functions in active learning in text classification and compare the performance with random sampling using 1% of the dataset size (Jacobs et al., 2021).

### 4.2 Experiment Results

Table 2: The result of data distillation

| Method | SST |
|---|---|
| Original Data | 0.9859 |
| Distilled Data | 0.8799 |
| Random Data | 0.6290 |

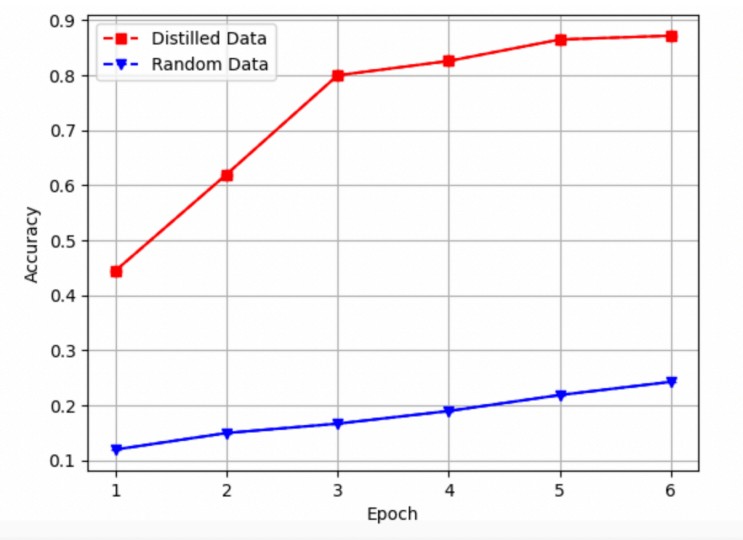

Figure 3: learning rate of data distillation

In this paper, 0.1% of the original training data are the number of samples in the random data and the distilled data. The result of data distillation in this experiment is shown as table2 and learning rate is shown as figure3.

The distilled data achieve about 88% accuracy and random data only achieve 63% accuracy. It is noticeable that although the distilled data cannot perform as well as the full data, the size of distilled data is much smaller than the full data with only 0.1% of the original data.

From the figure3 above, it is observed that training neural network on the distilled data is more easily than that on the random data. Looking at the red line which denotes epoch of distilled data rises fast to reach its highest level, while the blue line denoting the random data has a slowly increase. It means that data distillation is an effective way to train deep neural network in text classification.

Jacobs has discussed about various query functions in active learning in SST and the result is shown below (Jacobs et al., 2021). It is clear that on this dataset, active learning does not perform much better than random sampling. During 15 epochs, the gap between random sampling and active learning is really small. Therefore, active learning does not seem to have a significant performance in text classification.

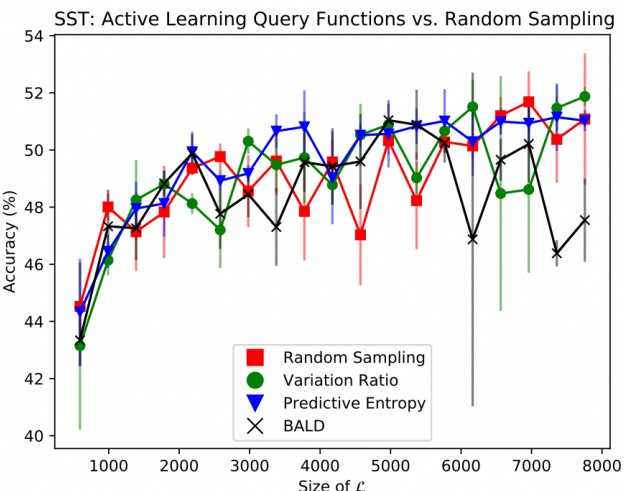

Figure 4: Active Learning

### 4.3 Data Analysis

It is noticeable that data distillation outperforms active learning in text classification with 88% accuracy of the original data. Here, we demonstrate the differences between active learning and data distillation.

(1)Algorithms:

The basic algorithm of data distillation is similar with knowledge distillation which is more like data compression using gradient descent. Thus, data distillation is able to maintain all information and knowledge from the original data. In addition, backpropagated gradient is the most effective method on data distillation. When training neural networks on a small number of data among a large number of data can achieve the optimal result, it is effective to use backpropagated gradient.

For active learning, it is necessary to compare various performances from different query functions. One of the most noticeable defects which compare with data distillation is that manual label leads to human disturb. Human labeled dataset in active learning is from true dataset distribution but not maintaining all information and knowledge from the true dataset. In addition, noise and outliers in large amounts of data will always be picked as the labeled dataset, which is bad for model training

(2)Hard to transfer:

As for active learning which is a data selecting strategy has difficulties in generality. It is not able to be generic among different tasks with various dataset distribution and labels, and in many occasions people need to spend more time in designing the specific active learning strategies for various human tasks.

Speaking of data distillation, it is a general method to reduce the dataset size. Among various industries, applying data distillation bases on a general way including selecting the distilled size of data and gradient descent. Therefore, when facing different dataset distribution and labels, data distillation play a significant role in generality.

(3)Unstable performance:

Data distillation focuses on transferring all knowledge and information from the original dataset to the distilled data using gradient descent to update parameters and find the most optimal distilled dataset which aims to minimize the loss function. Although the distilled data performs worse than the full original data, it does outdo the performance of active learning and shows the stable performance.

Compared with data distillation, active learning is used by its specified selection strategy, thus query function and the nature of the dataset are two main factors during the process. In practice, people need to first select and label data based on active learning, and if the strategy is not as good as random sampling at this point, they are not able to change or stop in time because the data has already been labeled and the sunk costs have already been incurred.

Active learning shows unstable performance among different datasets which own different distribution and labels, therefore when people want to solve human tasks in an efficient way, active learning is not as steady as data distillation.

## 5 Conclusion

In this paper, we explore data distillation which aims to distill all information and knowledge from the original data to a small number of synthetic dataset. We compare different performances between data distillation and active learning in text classification and discuss about discrepancies between two algorithms. The result is that the distilled data with only 0.1% of the original data achieve 88% accuracy, while active learning only performs 52% accuracy of the original data.

In conclusion, although data distillation has not been used in many industries, many surveys show the efficiency of data distillation in machine learning. We believe data distillation will play a significant role in machine learning and deep neural network in the future.

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
