# OpenReview forum: "A comparison of dataset distillation and active learning in text classification"
_ICLR.cc/2023/Conference — Submitted to ICLR 2023_

### Official Review · Reviewer_WRBC · 2022-10-20

**Confidence:** 5
**Clarity, Quality, Novelty And Reproducibility:** This paper is of low quality.
**Correctness:** 1
**Technical Novelty And Significance:** 1
**Empirical Novelty And Significance:** 1
**Recommendation:** 1

**Strength And Weaknesses:**

The research objective of this paper is neither clear nor reasonable. Experimental results of active learning are not reliable. The conclusion is not reasonable based on the simple experiments.

**Summary Of The Paper:**

This paper compares dataset distillation and active learning in text classification. Authors introduce basic methods and conduct experiments on the SST dataset.

**Summary Of The Review:**

This paper is below the standard of ICLR.

---

### Official Review · Reviewer_8K6w · 2022-10-22

**Confidence:** 4
**Clarity, Quality, Novelty And Reproducibility:** Data distillation and active learning…
**Correctness:** 4
**Technical Novelty And Significance:** 1
**Empirical Novelty And Significance:** 1
**Recommendation:** 1

**Strength And Weaknesses:**

Strength:
1. This is a good introductory article to compare Data Distillation and Active Learning.

Weakness:
1. This paper did not propose any new methodology or dataset.
2. There is only one evaluation dataset and two sets of experiments.

**Summary Of The Paper:**

This paper introduces and compares discrepancies of Data Distillation and Active Learning and evaluate them on NLP classification dataset: Stanford Sentiment Treebank.

**Summary Of The Review:**

This is a good introductory article to compare Data Distillation and Active Learning. But it did not propose any new methodology or dataset. And there are no new dataset proposed nor enough experiments to serve as benchmark.

---

### Official Review · Reviewer_wkGv · 2022-10-28

**Confidence:** 5
**Correctness:** 3
**Technical Novelty And Significance:** 1
**Empirical Novelty And Significance:** 1
**Recommendation:** 1

**Clarity, Quality, Novelty And Reproducibility:**

The paper contains a large number of English grammar errors, making it sometimes hard to read.

There is no novelty in the paper. Experiments are limited to one datasets.

I have no reason to doubt the reproducibility of the experimental results.

**Strength And Weaknesses:**

Strengths:
1. Good references to past papers that propose the ideas.
2. Clear introduction on the technical details.

Weaknesses:
1. No novel ideas are proposed in the paper.
2. Only one small dataset is used for experiments.
3. Discussions lack depth.

**Summary Of The Paper:**

The paper provides a comparison of data distillation and active learning for text classification. The models used in the paper are the most common types of distillation and active learning models. Experiments are performed on the ST dataset. Discussions are offered on the results.

**Summary Of The Review:**

No novelty. Strightforward comparison between distillation and active learning. Limited results and discussions.

---

### Decision · Program_Chairs · 2023-01-20

**Decision:**

Reject

**Justification For Why Not Higher Score:**

N/A

**Justification For Why Not Lower Score:**

N/A

**Metareview: Summary, Strengths And Weaknesses:**

This paper introduces a comparison of dataset distillation and active learning in text classification. Reviewers had numerous concerns about the paper but the authors did not respond to them. Therefore, I would like to reject the paper.